# Psychometric evaluation of the Chinese version of the Person-centred Climate Questionnaire - Staff version (PCQ-S)

Le Cai,[1] Gerd Ahlström,[2] Pingfen Tang,[1] Ke Ma,[3] David Edvardsson,[4] Lina Behm,[2] Haiyan Fu,[3] Jie Zhang,[3] Jiqun Yang[3]

[1]School of Public Health, Kunming Medical University, Kunming, China
[2]Department of Health Sciences, Faculty of Medicine, Lund University, Lund, Sweden
[3]Division of Palliative Care, The Third People's Hospital of Kunming, Kunming, China
[4]Austin Health/Northern Health Clinical Schools of Nursing, College of Science, Health and Engineering, La Trobe University, Bundoora, Victoria, Australia

**Correspondence to**
Dr Le Cai;
1018606825@qq.com

## ABSTRACT

**Objectives** The aim of the study was to evaluate the psychometric properties of a Chinese translation of the English version of the Person-centred Climate Questionnaire – Staff version (PCQ-S) for Chinese palliative care staff in a hospital context.

**Design** This was a cross-sectional design. The 14-item English PCQ-S was translated and backtranslated using established procedures. Construct validity and reliability including internal consistency and test-retest reliability were assessed among hospital staff. Construct validity was tested using principal component analysis (PCA), internal consistency was assessed using Cronbach's alpha, and test-retest reliability was evaluated with the weighted kappa (Kp), Pearson correlation coefficient (r) and intra-class correlation coefficient (ICC).

**Setting** This study was conducted in three hospitals in Kunming, the capital of Yunnan province in south-west China.

**Participants** A sample of hospital staff (n=163) on duty in the palliative care departments of three hospitals in Kunming consented to participate in the study.

**Results** The 14-item Chinese PCQ-S consists of the three subscales also present in other language versions. It showed strong internal consistency, with a Cronbach's alpha of 0.94 for the total scale, 0.87 for the safety subscale, 0.90 for the everydayness subscale and 0.88 for the community subscale. The Chinese PCQ-S had high test-retest reliability as evidenced by a high Kp coefficient and a high correlation coefficient for all scales between test and retest scores, on 'a climate of safety' (Kp=0.77, r=0.88, p<0.01), 'a climate of everydayness' (Kp=0.82, r=0.91, p<0.01), 'a climate of community' (Kp=0.75, r=0.79, p<0.01), and on overall scale scores (Kp=0.85, r=0.93, p<0.01). The ICC to evaluate the test-retest reliability was 0.97 (95% CI 0.95 to 0.98).

**Conclusions** The Chinese version of the PCQ-S showed satisfactory reliability and validity for assessing staff perceptions of person-centred care in Chinese hospital environments.

## INTRODUCTION

Population ageing is a global phenomenon and has become a significant public health problem worldwide. Along with high economic growth and demographic change over the last two decades, China also has one

### Strengths and limitations of this study

► This is the first study to validate the Person-centred Climate Questionnaire – Staff version (PCQ-S) in an Asian population.
► There was a high response rate (90%) in this study.
► The convenience sampling method used may limit ability to generalise the results.
► The Chinese PCQ-S has been tested only in this hospital palliative care environment.

of the most rapidly ageing populations in the world. The proportion of older people aged 60 years or more was 13.3% in 2010,[1] and is projected to reach 32.8% by 2050.[2] Studies have indicated that older people are more likely to have various conditions, particularly chronic disease and comorbidity which is difficult to treat.[3] As chronic diseases may result in disability in ageing patients, the rising number of older people increases the demand for hospitalisation and special care and support from multiple care professionals and providers.[4] This presents a key challenge for Chinese healthcare systems to provide high-quality care for this group. Rapid population ageing in China is also increasing the numbers of older people who are likely to require palliative care as a result of higher levels of poor health and chronic disease. However, palliative care in China has developed more slowly than in high-income Western countries. Palliative care is based on harmony between the mind and body in traditional Chinese medicine (TCM) with its long history developed over a few thousand years.[5 6]

In recent decades, person-centred care (PCC) has been recognised as a focus of quality elderly care services, which emphasises the individual's perspective and active participation in the care process.[7] PCC as a concept implies assisting an individual in various ways to be a 'whole' human being

by encouraging them to participate in decisions and adjusting the physical environment and the type of care to fit the needs of each patient. It is defined as 'valuing people as individuals' when delivering healthcare[8] and is based on people's subjective experience of illness instead of the disease itself.[9–11] The care process becomes the foundation for how PCC should be provided and the quality of the relationship between the professional caregiver and the care recipient is key.[12–14] Person-centredness is now regarded as a central feature of high-quality long-term care for older persons. As such, PCC must become a priority for the care organisation, and the system needs to support and sustain this through policy and procedures, job descriptions and education.[15] PCC improves autonomy in the elderly in care through its focus on individual care plans and support for next of kin, who are seen as important resources.[16] There is evidence to indicate that the person-centredness of a setting is associated with staff satisfaction with work,[17] as staff perceptions of and relationships with patients are crucial for care quality. Also, for the older person, a person-centred setting has been shown to increase well-being and decrease discomfort.[18 19]

Internationally, various instruments have been developed to evaluate the PCC perspectives of professionals who work in elderly care facilities, including the Person-centred Climate Questionnaire - Staff version (PCQ-S),[20] the Person-centred Care Assessment Tool (P-CAT),[21] the Staff Assessment Person Directed Care (PDC) measure,[22] individualised care (IC),[23] and the Staff Person-Centred Practices in Assisted Living (Staff PC-PAL) questionnaire.[24]

Edvardsson and colleagues developed the Swedish-language Person-centred Climate Questionnaire – Staff version (PCQ-S) for evaluating the extent to which the climate of care environments is experienced as being person-centred by staff.[20 25] The questionnaire comprises three subscales (safety, everydayness and community). It has been validated with older persons being cared for in hospitals, and been shown to have satisfactory psychometric properties, with an overall Cronbach's alpha of 0.88 and values of 0.84, 0.80 and 0.77, respectively, for the three subscales, and satisfactory test–retest reliability, with an average intra-class correlation coefficient (ICC) of 0.51 (95% CI 0.47 to 0.75). It is commonly used internationally and has been translated from Swedish into Norwegian[26] and English,[27] and the English version has also been translated into Slovenian.[28] Both the original and the translated Norwegian, English and Slovenian scales have been demonstrated to be valid and reliable tools for assessing staff perceptions of person-centredness. However, there is no Chinese version of the PCQ-S, which presents a barrier to measuring and developing person-centred care and to conducting further studies in China and making international comparisons. We believe the English PCQ-S was the most suitable of the different versions for adaptation to the Chinese context as it was the closest to Chinese attitudes. So the purpose of this study was to adapt the English version of the PCQ-S for Chinese healthcare staff and to evaluate the psychometric properties of the translated Chinese version in a hospital palliative care context.

## METHODS

### Instrument

The English PCQ-S questionnaire consists of 14 items and has three subscales (a climate of safety, everydayness and community).[25] A climate of safety is measured through items 1–5, everydayness through items 6–10, and community through items 11–14. Scoring is performed on a 6-point Likert-type scale, ranging from 0 (No, I disagree completely) to 5 (Yes, I agree completely). Aggregated scores are calculated using simple sum scores at subscale and total scale levels, and range from 0 to 70, with higher scores indicating a setting perceived as being more person-centred. The English PCQ-S has previously been used and tested in hospital settings, and has been shown to be a valid and reliable tool for assessing staff perceptions of the unit's person-centredness.[27]

### Translation and cross-cultural adaptation of the PCQ-S

Translation and cross-cultural adaptation was carried out according to previously published international test commission guidelines.[29 30] First, forward translation from English to Chinese was performed independently by three native Chinese speakers, two of whom were university graduates with a public health background while the other was a physician familiar with palliative care. A consensus version was obtained after discussion between the three translators. Second, the consensus version was back-translated into English by two translators blinded to the procedures of the forward translation. However, the back-translated version was not discussed with the authors of the English-language version of the PCQ-S. Finally, a thorough comparison of the original, translated and back-translated versions was conducted by an expert committee, which consisted of all translators, three palliative care physicians and two university professors. Discrepancies in translations were discussed and resolved, some wording was adapted to the Chinese cultural setting, and a consensus pre-final version was established. A final Chinese version was generated through face validity after the pre-final version was pre-tested on 10 staff from a municipal hospital in Kunming. No changes were made after the pre-testing. The 10 staff participating in face validity of the pre-final version did not subsequently take part in the study itself.

### Sample and participants

Three municipal hospitals in Kunming, the capital of Yunnan province in south-west China, were selected using a convenience sampling method. The following inclusion criterion was used: a municipal hospital in Kunming city with a department of palliative care. Participation was approved by the hospital directors. All staff (n=182) on

duty (on both the morning and afternoon shifts on one specific day) in the departments of palliative care in these three hospitals were considered eligible for participation and invited to complete the Chinese PCQ-S questionnaire. The sample size in our study was in accordance with the criteria proposed by Terwee *et al*.[31] Eligible staff received both oral and written information about the study. Before data collection, each participant was given a full explanation of the purpose of the research, and was informed that they were under no obligation to participate in the study and could withdraw from the study at any time without any prejudice or repercussions. A total of 163 agreed to participate, representing an overall response rate of 90%. The participants completed questionnaires for both the test and retest assessments.

### Data collection

Demographic data were collected along with the questionnaire answers and included staff age, gender, level of education, duration of work experience, ethnicity and healthcare staff position. Each participant was assigned a number by the data collector to indicate his or her identity, so they were anonymous with regard to completing the questionnaire. Two university graduates distributed questionnaires to all participants, and completed questionnaires were anonymously collected on site. To examine test-retest reliability, all participants were asked to complete the same PCQ-S questionnaire 1 week later. Those unavailable on that day were invited to complete the PCQ-S questionnaire on the soonest possible date. Data were collected during October and November 2016.

### Psychometric evaluation

No variable had missing values. All complete data were included in the analysis. Construct validity was estimated using exploratory factor analysis (principal component analysis, PCA) with both varimax orthogonal and oblique orthogonal rotation, and goodness-of-fit through confirmatory factor analysis.[32] The analysis indicated no difference between the two methods, so only the results from the analysis with varimax orthogonal rotation are presented.

Bartlett's test of sphericity was used to assess whether the correlation between items was adequate based on a criterion of p<0.0001. The Kaiser-Meyer-Olkin (KMO) statistic was used to measure sample adequacy based on a criterion of ≥0.7. Principal components were extracted when Kaiser's criterion of eigenvalues was ≥1. A component loading cut-off of 0.5 was used to decide if an item loaded on a specific component.[33] PCA with oblique rotation was performed to ensure independence of the items.

Reliability testing included assessments of internal consistency and test-retest reliability. Internal consistency for total and subscale scores was estimated using the Cronbach's alpha coefficient, and the cut-off scores for acceptable reliability were set to item-total correlations of ≥0.5 and in such a way that Cronbach's alpha would not be increased by item deletion.[34] A Cronbach's alpha

**Table 1** Demographic characteristics of the study sample (n=163)

| Characteristics | n (%) |
|---|---|
| Gender | |
| Female | 151 (92.6) |
| Male | 12 (7.4) |
| Age (years) | |
| 18–30 | 95 (58.3) |
| 31–39 | 33 (20.2) |
| ≥40 | 35 (21.5) |
| Level of education | |
| High school | 7 (4.3) |
| Secondary school | 37 (22.7) |
| Junior college | 65 (39.9) |
| Bachelor degree or higher | 54 (33.2) |
| Ethnicity | |
| Han | 118 (72.4) |
| Minorities | 45 (27.6) |
| Healthcare staff | |
| Registered nurse | 101 (62.0) |
| Enrolled nurse | 29 (17.8) |
| Physician | 33 (20.2) |

between >0.8 and >0.95 was taken to indicate that the questionnaire had good or excellent internal consistency.[34] Test–retest reliability was evaluated through the weighted kappa coefficient (Kp), Pearson correlation coefficient (r) and a single measure two-way mixed effects model ICC, where an ICC >0.80 was taken to indicate satisfactory reliability.[35] The paired t-test was used to determine whether the mean scores of the test and retest questionnaires differed significantly. All statistical significance decisions were based on two-tailed p values of <0.05. All data analyses were conducted using SPSS 17.0 software.

## RESULTS

### Demographic characteristics of the study group

Table 1 shows the demographic characteristics of the study group. The sample consisted of 92.6% female and 7.4% male staff. The mean±SD age was 31.6±10.1 years, with an average length of work experience in healthcare of 8.1±7.4 years. More than a quarter of respondents belonged to ethnic minorities. Most participants were registered nurses (62.0%) or enrolled nurses (17.8%). About one third (33.2%) of the participants had a Bachelor's degree or higher (see table 1).

### Construct validity

The results of the PCA with Bartlett's test (p<0.0001) and the KMO measure (0.91) indicated that correlations between items were sufficiently large to perform the PCA. Only the first three components had eigenvalues greater

**Table 2** Rotated component matrix for PCA of the Chinese PCQ-S (n=163)

| Item number | Item content | Factor loadings | | |
| --- | --- | --- | --- | --- |
| | | Subscale 1: A climate of safety | Subscale 2: A climate of everydayness | Subscale 3: A climate of community |
| 1 | A place where I feel welcome | 0.83 | | |
| 2 | A place where I feel acknowledged as a person | 0.84 | | |
| 3 | A place where I feel I can be myself | 0.58 | | |
| 4 | A place where the patients are in safe hands | 0.66 | | |
| 5 | A place where the staff use a language that the patients can understand | 0.60 | | |
| 6 | A place which feels homely even though it is in an institution | | 0.82 | |
| 7 | A place where there is something nice to look at | | 0.81 | |
| 8 | A place where it is quiet and peaceful | | 0.78 | |
| 9 | A place where it is possible to get unpleasant thoughts out of your head | | 0.74 | |
| 10 | A place which is neat and clean | | 0.68 | |
| 11 | A place where it is easy for the patients to keep in contact with their loved ones | | | 0.64 |
| 12 | A place where it is easy for the patients to receive visitors | | | 0.87 |
| 13 | A place where it is easy for the patients to talk to the staff | | | 0.85 |
| 14 | A place where the patients have someone to talk to if they so wish | | | 0.66 |
| Total variance explained (%) | 73.3 (total three subscales) | 55.6 | 9.5 | 8.2 |
| Cronbach's alpha | 0.94 (total 14 items) | 0.87 | 0.90 | 0.88 |

PCA, principal component analysis; PCQ-S, Person-centred Climate Questionnaire – Staff version.

than 1, explaining 73.3% of the total variance. Therefore, the PCA resulted in a three-component rotated solution. As shown in table 2, the first and the second components consisted of five items (loadings between 0.58 and 0.84 vs loadings between 0.68 and 0.82), where the first component confirmed the subscale 'a climate of safety' and where the second component confirmed the subscale 'a climate of everydayness' in the setting. The third component comprised four items (loadings between 0.64 and 0.87), and confirmed the subscale 'A climate of community'.

The three-component model was also evaluated by confirmative factor analysis, and goodness-of-fit was estimated using indices of the root mean square error of approximation (RMSEA), the normed fit index (NFI) and the comparative fit index (CFI). The results indicated that the goodness-of-fit of the questionnaire was 0.78 for the RMSEA, 0.91 for the NFI and 0.92 for the CFI. Thus, the confirmatory factor analysis supported the exploratory findings, and the three-component model provided adequate fit indices for the questionnaire.

### Reliability

Table 2 shows that the Cronbach's alpha coefficient of the 14-item Chinese PCQ-S was 0.94 for the total scale,

0.87 for the safety subscale, 0.90 for the everydayness subscale, and 0.88 for the community subscale, indicating strong internal consistency and reliability overall. Furthermore, the corrected item-total correlations for all items ranged from 0.54 to 0.79, indicating that each item correlated adequately with the total score and thus that the scale is homogenous without any item being redundant (table 3).

Table 4 presents the results from the test-retest reliability assessment of the Chinese PCQ-S. The Kp statistic for the overall scale scores was 0.85 ($p<0.001$), indicating that the Chinese PCQ-S instrument has substantial reliability. For each subscale, the results varied from 0.75 to 0.82 ($p<0.001$). According to the Pearson's correlation coefficient analyses, the Chinese PCQ-S demonstrated high correlation between test and retest on all scale levels: on the subscales 'a climate of safety' ($r=0.88$, $p<0.01$), 'a climate of everydayness' ($r=0.91$, $p<0.01$) and 'a climate of community' ($r=0.79$, $p<0.01$), as well as on the overall scale scores between test and retest ($r=0.93$, $p<0.01$). A paired t-test also confirmed that there was no significant difference between the mean scores of the PCQ-S at regarding the test and retest values ($p>0.05$). The ICC of the total score between test and retest was 0.97, providing

**Table 3** Item performance and reliability testing of the Chinese Person-centred Climate Questionnaire – Staff version (PCQ-S) (n=163)

| Item number | Item content | Mean±SD | Corrected item: total correction | Cronbach's alpha if item deleted |
|---|---|---|---|---|
| 1 | A place where I feel welcome | 4.04±0.93 | 0.62 | 0.93 |
| 2 | A place where I feel acknowledged as a person | 4.07±0.92 | 0.54 | 0.93 |
| 3 | A place where I feel I can be myself | 3.58±1.32 | 0.70 | 0.93 |
| 4 | A place where the patients are in safe hands | 4.06±0.96 | 0.72 | 0.93 |
| 5 | A place where the staff use a language that the patients can understand | 3.90±1.01 | 0.72 | 0.93 |
| 6 | A place which feels homely even though it is in an institution | 3.80±1.13 | 0.77 | 0.93 |
| 7 | A place where there is something nice to look at | 3.60±1.15 | 0.76 | 0.93 |
| 8 | A place where it is quiet and peaceful | 3.80±1.04 | 0.78 | 0.93 |
| 9 | A place where it is possible to get unpleasant thoughts out of your head | 3.20±1.34 | 0.66 | 0.93 |
| 10 | A place which is neat and clean | 3.85±1.01 | 0.70 | 0.93 |
| 11 | A place where it is easy for the patients to keep in contact with their loved ones | 3.88±1.03 | 0.79 | 0.93 |
| 12 | A place where it is easy for the patients to receive visitors | 3.40±1.36 | 0.59 | 0.93 |
| 13 | A place where it is easy for the patients to talk to the staff | 3.72±1.16 | 0.71 | 0.93 |
| 14 | A place where the patients have someone to talk to if they so wish | 3.94±1.03 | 0.67 | 0.93 |

further support that the scale had satisfactory test–retest reliability.

## DISCUSSION

This is the first study to validate the PCQ-S in an Asian population. The results of the study indicated that the cross-culturally adapted Chinese version of the PCQ-S showed excellent reliability and validity for evaluating staff perceptions of person-centredness in Chinese hospital contexts, which will enable further studies and international comparisons.

In this study, the English PCQ-S was translated and cross-culturally adapted for use in a Chinese setting and showed satisfactory psychometric properties (construct validity, test-retest reliability and internal consistency). During our translation of the English PCQ-S into Chinese, a minor cultural discrepancy was encountered and one item of the PCQ-S was modified accordingly: 'peaceful' was replaced with 'harmonious' as this word is closer to Chinese culture. Construct validity was estimated using PCA with varimax orthogonal rotation, resulting in a stable three-factor solution explaining 73.3% of the total variance. The ICC for the overall Chinese PCQ-S scale was 0.97, and for the three subscales was 0.93, 0.95 and 0.92, respectively, demonstrating that the test-retest reliability of the overall scale and different domains was excellent. Furthermore, strong internal consistency of the Chinese PCQ-S was demonstrated, as evidenced by a Cronbach's alpha of 0.89 for the total scale, 0.87 for the safety subscale, 0.90 for the everydayness subscale and 0.88 for the community subscale.

**Table 4** Test-retest reliability of the Chinese Person-centred Climate Questionnaire – Staff version (PCQ-S) (n=163)

| Scale dimension | First test (mean±SD) | Second test (mean±SD) | p Value | Weighted kappa (Kp) | Pearson correlation coefficient (r) | ICC (95% CI) |
|---|---|---|---|---|---|---|
| A climate of safety | 19.7±4.2 | 19.8±4.0 | 0.30 | 0.77 | 0.88 | 0.93 (0.91 to 0.95) |
| A climate of everydayness | 18.3±4.8 | 18.1±4.9 | 0.38 | 0.82 | 0.91 | 0.95 (0.93 to 0.96) |
| A climate of community | 15.0±4.0 | 14.7±4.1 | 0.18 | 0.75 | 0.79 | 0.92 (0.89 to 0.94) |
| Overall scale | 52.9±11.4 | 52.6±11.7 | 0.40 | 0.85 | 0.93 | 0.97 (0.95 to 0.98) |

This Chinese version of the PCQ-S had the same subscale structure as the Swedish, Norwegian and Slovenian versions, as it contained three subscales (a climate of safety, everydayness and community) consisting of 14 items. However, it had a different structure to the English PCQ-S which has four subscales (a climate of safety, everydayness, community and comprehensibility) consisting of 14 items, which may reflect a difference in cultural context. Even though the original English PCQ-S had a slightly different structure as it has four subscales, the instrument developers have recently suggested using versions with three subscales for scoring and comparison purposes.[20]

In the Chinese PCQ-S, the ICC (0.97) and Cronbach's alpha for the total scale (0.94) were much higher than in the Swedish (0.51 vs 0.88) and English (0.75 vs 0.89) versions, and Cronbach's alpha for the total scale was also higher than in the Norwegian version (0.92). Due to the larger the sample size in our study, which differs from those in the above studies, the Chinese PCQ-S may have stronger test-retest reliability and internal consistency compared with other language versions of the PCQ-S. The results demonstrated that the Chinese PCQ-S has good reproducibility and maintains the properties of the original version, and can thus be used in Chinese hospital environments.

The following limitations of the present study should be noted. First, the study employed a convenience sampling method to select staff working in palliative care in public hospitals, which may limit the generalisability of the results to staff in general in Chinese hospitals or to staff working in other healthcare settings. Second, the Chinese PCQ-S questionnaire has been tested only in this hospital environment, so further psychometric testing in other settings, such as nursing homes, would be helpful for further comparison of the Chinese PCQ-S in different contexts and settings. Third, the questionnaire was translated only from the secondary English version, not from the original Swedish version. Fourth, with respect to psychometric assessment of the PCQ-S, criterion-related validity, convergent validity and discriminative validity were not taken into account. Further study is needed to explore this in the future. Fifth, the back-translated version was not validated because of the cross-cultural adaptation.

## CONCLUSION

The 14-item Chinese PCQ-S is a cross-culturally adapted version of the English PCQ-S and showed excellent psychometric properties in terms of reliability and validity for evaluating staff perceptions of the person-centredness in Chinese hospital environments. Our results indicated that the Chinese version of the PCQ-S can be utilised for the future measurement and development of person-centred care in China and for conducting cross-cultural international comparisons with, for example, Sweden.

**Acknowledgements** We would like to thank Ms Huang Jingjing and Ms Wu Chao, two postgraduate students in the School of Public Health, Kunming Medical University, who helped us with data collection and Magnus Persson, Lund University, Sweden for providing comments on the paper.

**Contributors** LC was responsible for the study design, data analysis, and drafting the paper. GA , PT and KM contributed to the study design and provided comments on the paper during the writing process. DE and LB provided comments on the paper during the writing process. HF, JZ and JY were responsible for the data collection.

**Funding** The study was supported by grants from the National Natural Science Fund of China (Grant number: 81611130077), the Swedish Research Council (Grant number: 2015?06243) and the Major Union Specific Project Foundation of Yunnan Provincial Science and Technology Department and Kunming Medical University (2016).

**Competing interests** None declared.

**Ethics approval** This study was approved by the Ethics Committee of Kunming Medical University.

**Provenance and peer review** Not commissioned; externally peer reviewed.

**Data sharing statement** The datasets used and/or analysed during the current study is available from the corresponding author on reasonable request.

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
