## [Reviewer comments · BMJ Open]

ARTICLE DETAILS

TITLE (PROVISIONAL)	Psychometric evaluation of the Chinese version of the Person-centred Climate Questionnaire for staff
AUTHORS	Cai, Le; Ahlström, Gerd; Tang, Pingfen; Ma, Ke; Edvardsson, David; Behm, Lina; Fu, Haiyan; Zhang, Jie; Yang, Jiqun

VERSION 1 - REVIEW

REVIEWER	Mariusz Panczyk, PhD Division of Teaching and Outcomes of Education, Faculty of Health Science, Medical University of Warsaw, POLAND
REVIEW RETURNED	23-Apr-2017

GENERAL COMMENTS	The manuscript presented for evaluation is focused on cultural adaptation and psychometric validation of the Person-centred Climate Questionnaire – staff version (PCQ-S). The implementation of the idea of person-centred care helps to increase the quality of patient care and improves patient safety. The scale presented in the manuscript is a possible instrument that could be used to perform measurements of to what extent staff perceive different clinical settings or climates as being person-centred, an attribute often described as an essential component of quality health care. These are a few comments that would be helpful in improving the manuscript before the editors of the BMJ Open consider whether to publish it. 1. Introduction The first paragraph reviewed the present demographic situation and outlined a picture of trends observed in Chinese society, hence providing a view of challenges faced by the Chinese healthcare system. The authors pointed to the introduction of the concept of person-centred care into the patient care system as an essential component of necessary reforms. This was supported by a number of available publications (e.g. McCormack B et al., Edvardsson D et al.) and seemed to be sufficient justification to move towards the implementation of the idea of person-centred care in China as well. Unfortunately, it seems that the authors provided insufficient justification for using the PCQ-S as the assessment tool. There was no information whether this was the only available questionnaire/scale relating to the person-centred care. In addition, it would be appropriate to devote a few words to theoretical assumptions that were the cornerstone of the original Swedish version of the PCQ-S and explain to what extent, according to the authors, these assumptions were accurate in relation to the Chinese culture. Is it possible that the idea of person-centred care is equally understood regardless of the specific nature and conditions in different countries?
--

2. Instrument

[page 5, line 56] "The English PCQ-S has previously been used and tested in hospital settings, and demonstrated to be a valid and reliable tool for assessing staff perceptions of the unit person-centredness [20]". Is there only one publication referring to the validation of the English-language version of the PCQ-S? Would it be advisable to mention psychometric properties of the original questionnaire (Swedish-language version), that is a prototype for the later English-language version?

3. Translation and cross-cultural adaptation of the PCQ-S

Despite the fact that the authors referred to two types of guidelines concerning the principles of psychometric validation, no reasons were provided why only the English-language version was used for validation. Good practices show that in the case of numerous language versions, a translation should be provided from different sources (different languages), and then a consensus version should be developed. This is all the more important because the original language version is Swedish, not English.

Was the back-translated version consulted with the authors of the English-language version of the PCQ-S? Such consultation increases validity of the translation and allows for eliminating any inaccuracies from the new version regarding for instance the selection of expressions or translation of ambiguities.

[page 6, line 28] "A few wordings were adapted to the Chinese cultural setting (...)". Since the manuscript deals with cultural adaptation, it seems that the outcomes of this particular validation phase should be somehow made available to readers.

Did 10 persons participating in face-validity of the pre-final version subsequently take part in the study itself?

4. Data collection

[page 7, line 7] "(...) completed questionnaires were anonymously collected on site." If the study was anonymous, how were the data obtained in the test phase linked with the data obtained in the retest phase? Both phases were separated by one week.

5. Psychometric evaluation

PCA with varimax orthogonal rotation was used for the assessment of the construct validity. Assuming, however, that particular subscales may be strongly correlated, it would be advisable to use an oblique rotation method. In addition, would it not be appropriate to use a confirmatory factor analysis (CFA)? The validation of the original language version of PCQ-S demonstrated that there were three subscales. It would be useful to assess the consistency of the assumptions of such a model with the validation results of the Chinese version. In addition, the authors provided no reference concerning the conditions applied for carrying out PCA (EFA?).

When assessing the internal consistency, the authors did not provide a satisfactory level of Cronbach's alpha coefficient.

With respect to test-retest reliability, the intraclass correlation (ICC) was measured correctly. However, for the subscales scores, Bland Altman 95% limits of agreement of differences were determined, as a measure of the upper and lower limits of the differences between the scores on the two occasions of testing (see details: Giavarina D. Understanding Bland Altman analysis. *Biochem Med (Zagreb)*. 2015 Jun 5;25(2):141-51. doi: 10.11613/BM.2015.015).

6. Results: Construct validity

The authors did not explain how the factor solution obtained in the

study met the Kaiser's criterion. All that is known is that a 3-factor solution explained a total of 73.3% of the total variance. Did only the first three eigenvalues meet the criterion (≥ 1)?

7. Results: Reliability

[page 8, line 51] "(...) each item correlated adequately with the total score and thus that the scale is homogenous without any item being redundant". The EFA method needs to be used for the assessment of one-dimensionality of subscales (see details: Evaluating Psychometric Properties: Dimensionality and Reliability [In:] Furr, Mike. Scale construction and psychometrics for social and personality psychology. SAGE Publications Ltd, 2011).

8. Discussion

[page 9, line 28]: "In the Chinese PCQ-S, the ICC (0.97) and Cronbach's alpha for the total scale (0.94) were much higher than those recorded in Swedish (0.51 vs. 0.88) and English (0.75 vs. 0.89) versions, and the Cronbach's alpha for the total scale was also higher than the Norwegian version (0.92), (...)". It is important to recognize that the value of Cronbach's alpha coefficient is not only correlated with the internal consistency, but also with the sample size. Therefore, while presenting results in this area, it is important to bear in mind the comparability of groups with respect to the sample size in particular validation studies.

[page 10, line 45] With reference to the limitations of the study, it would also be worth mentioning that the questionnaire had been translated only from the secondary English version. Moreover, with respect to the psychometric assessment of the PCQ-S, convergent validity and discriminative validity were not taken into account. Additionally, the back-translated version was not validated, which, of course, would cause some technical difficulties due to the necessity to conduct the study among bilingual persons, but, at the same time, this would significantly increase confidence in the quality of cultural adaptation.

Summary

The manuscript presented for evaluation concerns an important issue of assessing the possibility of using the concept of person-centred care. This is important because the emerging negative demographic trends call for new solutions in patient care in numerous world countries, including China. The present PCQ-S validation outcomes may later contribute to an increase in the range of applications of standardised tools used for planning changes and implementing the concept of person-centred care in Chinese healthcare centres and perhaps in other Asian countries as well. Making improvements accordingly will raise the quality of the texts itself and provide readers with a better understanding of the principles of a proper validation of scales and questionnaires from a different cultural background.

The manuscript presented for evaluation is focused on cultural adaptation and psychometric validation of the Person-centred Climate Questionnaire – staff version (PCQ-S). The implementation of the idea of person-centred care helps to increase the quality of patient care and improves patient safety. The scale presented in the manuscript is a possible instrument that could be used to perform measurements of to what extent staff perceive different clinical settings or climates as being person-centred, an attribute often described as an essential component of quality health care. These are a few comments that would be helpful in improving the manuscript before the editors of the BMJ Open consider whether to

publish it.

1. Introduction

The first paragraph reviewed the present demographic situation and outlined a picture of trends observed in Chinese society, hence providing a view of challenges faced by the Chinese healthcare system. The authors pointed to the introduction of the concept of person-centred care into the patient care system as an essential component of necessary reforms. This was supported by a number of available publications (e.g. McCormack B et al., Edvardsson D et al.) and seemed to be sufficient justification to move towards the implementation of the idea of person-centred care in China as well. Unfortunately, it seems that the authors provided insufficient justification for using the PCQ-S as the assessment tool. There was no information whether this was the only available questionnaire/scale relating to the person-centred care. In addition, it would be appropriate to devote a few words to theoretical assumptions that were the cornerstone of the original Swedish version of the PCQ-S and explain to what extent, according to the authors, these assumptions were accurate in relation to the Chinese culture. Is it possible that the idea of person-centred care is equally understood regardless of the specific nature and conditions in different countries?

2. Instrument

[page 5, line 56] "The English PCQ-S has previously been used and tested in hospital settings, and demonstrated to be a valid and reliable tool for assessing staff perceptions of the unit person-centredness [20]". Is there only one publication referring to the validation of the English-language version of the PCQ-S? Would it be advisable to mention psychometric properties of the original questionnaire (Swedish-language version), that is a prototype for the later English-language version?

3. Translation and cross-cultural adaptation of the PCQ-S

Despite the fact that the authors referred to two types of guidelines concerning the principles of psychometric validation, no reasons were provided why only the English-language version was used for validation. Good practices show that in the case of numerous language versions, a translation should be provided from different sources (different languages), and then a consensus version should be developed. This is all the more important because the original language version is Swedish, not English.

Was the back-translated version consulted with the authors of the English-language version of the PCQ-S? Such consultation increases validity of the translation and allows for eliminating any inaccuracies from the new version regarding for instance the selection of expressions or translation of ambiguities.

[page 6, line 28] "A few wordings were adapted to the Chinese cultural setting (...)". Since the manuscript deals with cultural adaptation, it seems that the outcomes of this particular validation phase should be somehow made available to readers.

Did 10 persons participating in face-validity of the pre-final version subsequently take part in the study itself?

4. Data collection

[page 7, line 7] "(...) completed questionnaires were anonymously collected on site." If the study was anonymous, how were the data obtained in the test phase linked with the data obtained in the retest phase? Both phases were separated by one week.

5. Psychometric evaluation

PCA with varimax orthogonal rotation was used for the assessment of the construct validity. Assuming, however, that particular subscales may be strongly correlated, it would be advisable to use

an oblique rotation method. In addition, would it not be appropriate to use a confirmatory factor analysis (CFA)? The validation of the original language version of PCQ-S demonstrated that there were three subscales. It would be useful to assess the consistency of the assumptions of such a model with the validation results of the Chinese version. In addition, the authors provided no reference concerning the conditions applied for carrying out PCA (EFA?). When assessing the internal consistency, the authors did not provide a satisfactory level of Cronbach's alpha coefficient. With respect to test-retest reliability, the intraclass correlation (ICC) was measured correctly. However, for the subscales scores, Bland Altman 95% limits of agreement of differences were determined, as a measure of the upper and lower limits of the differences between the scores on the two occasions of testing (see details: Giavarina D. Understanding Bland Altman analysis. *Biochem Med (Zagreb)*. 2015 Jun 5;25(2):141-51. doi: 10.11613/BM.2015.015).

6. Results: Construct validity

The authors did not explain how the factor solution obtained in the study met the Kaiser's criterion. All that is known is that a 3-factor solution explained a total of 73.3% of the total variance. Did only the first three eigenvalues meet the criterion (≥ 1)?

7. Results: Reliability

[page 8, line 51] "(...) each item correlated adequately with the total score and thus that the scale is homogenous without any item being redundant". The EFA method needs to be used for the assessment of one-dimensionality of subscales (see details: Evaluating Psychometric Properties: Dimensionality and Reliability [In:] Furr, Mike. Scale construction and psychometrics for social and personality psychology. SAGE Publications Ltd, 2011).

8. Discussion

[page 9, line 28]: "In the Chinese PCQ-S, the ICC (0.97) and Cronbach's alpha for the total scale (0.94) were much higher than those recorded in Swedish (0.51 vs. 0.88) and English (0.75 vs. 0.89) versions, and the Cronbach's alpha for the total scale was also higher than the Norwegian version (0.92), (...)". It is important to recognize that the value of Cronbach's alpha coefficient is not only correlated with the internal consistency, but also with the sample size. Therefore, while presenting results in this area, it is important to bear in mind the comparability of groups with respect to the sample size in particular validation studies.

[page 10, line 45] With reference to the limitations of the study, it would also be worth mentioning that the questionnaire had been translated only from the secondary English version. Moreover, with respect to the psychometric assessment of the PCQ-S, convergent validity and discriminative validity were not taken into account. Additionally, the back-translated version was not validated, which, of course, would cause some technical difficulties due to the necessity to conduct the study among bilingual persons, but, at the same time, this would significantly increase confidence in the quality of cultural adaptation.

Summary

The manuscript presented for evaluation concerns an important issue of assessing the possibility of using the concept of person-centred care. This is important because the emerging negative demographic trends call for new solutions in patient care in numerous world countries, including China. The present PCQ-S validation outcomes may later contribute to an increase in the range of applications of standardised tools used for planning changes and

	implementing the concept of person-centred care in Chinese healthcare centres and perhaps in other Asian countries as well. Making improvements accordingly will raise the quality of the texts itself and provide readers with a better understanding of the principles of a proper validation of scales and questionnaires from a different cultural background.
--	--

REVIEWER	Rie Chiba University of Hyogo Japan
REVIEW RETURNED	10-May-2017

GENERAL COMMENTS	Thank you for the opportunity to review the paper entitled “Psychometric evaluation of the Chinese version Person-centred Climate Questionnaire – staff.” In this paper, the authors examined its reliability and validity of the scale in the palliative ward settings. The article is well written and interesting. I think it is valuable to be published in the journal. I hope some points described below would be the tips to refine the article. 1. Background In this paper, the authors conducted the survey only in palliative wards. Thus the explanation about general circumstances of palliative care in China, as well as the significance about the assessment of person-centered climate among the staff in palliative words may be needed. 2. Sample and Participants Though it is mentioned that only oral information about the study was provided, did they receive writing information? If so, the explanation may be added. 3. Sample and Participants The response rate (90%) seems considerably high. And it also seems too impeccable that there was no missing data among 163 participants at two time points. Thus the authors may want to describe about them including whether the explanation about such as the non-participants would not be disadvantaged, or participants were allowed to withdraw from the study at any time without prejudice were provided or not. 4. Sample and Participants Additional explanation about the timeframe of the test-retest may be desirable because there might be some participants who could not answer the retest survey just after a week from the baseline survey, because of their work schedule. 5. Psychometric evaluation The authors stated that they chose varimax rotation to ensure independence of the items. But I wonder the correlations among each items might be hypothesized. Thus more convincing explanation including the description about the rotations in the earlier studies may be desirable if the rotation was adopted. 6. Psychometric evaluation
---

	I wonder why the criterion-related validity was not examined in this study. If such data was obtained, please consider to show the result of the analyses. 7. Reliability Regarding test-retest reliability, the authors may want to analyze using weighted kappa which would show more rigorous result compared to the Pearson's correlation. 8. Discussion The first paragraph except for the last sentence may be better to be deleted or removed to Introduction section. In addition, the second section also seems duplicate the Introduction. Briefer summary of the result of the current study seems preferable. 9. Discussion In relation to the third paragraph, more concrete description of the factorial constructions in the earlier studies and profound discussion may add some implication of the study, including the difference of the culture between China and Western countries. 10. Limitation The author may want to add the limitation that criterion-related validity was not examined in this study.
--	---

REVIEWER	Dominika Vrbnjak University of Maribor Faculty of Health Sciences
REVIEW RETURNED	15-May-2017

GENERAL COMMENTS	Thank you for an interesting manuscript, it contains useful content. The manuscript is well written and adds to the international literature on measuring person-centeredness of environments. However, there are some issues to be considered, that might improve the manuscript further.  1. Study objective is clearly defined. 2. Abstract is balanced and complete. The sample included health care staff only from palliative care, this should be evident also from abstract and abstract summary. 3. The study design is adequate to address the study objective. 4. Methods are sufficiently described to allow the study to be repeated. But, no sample size calculation is available. KMO was performed to measure the sampling adequacy, however, this was done after distributing the questionnaires. Justifying the rationale for sample size and convenience sampling is needed. Justifying the choice of three hospitals would also be appropriate. Also, it would be useful if a more detailed description of what all staff on duty is meant (morning shift only?). In Data collection section, page 7, line 3-5 authors describe collecting demographic data. In addition to listed age, sex, level of education and duration of work experience, the authors have also collected data about ethnicity and health care staff position, this should also be listed here. 5. Ethics approval is stated; however, there could be more in-depth description of ethical issues, especially as test-retest was done.
---

Authors could explain how anonymity was assured and describe coding of the questionnaire if this was done.

6. Outcomes are clearly defined.

7. Statistical methods adequately match the study. Appropriate statistical references could be included on page 7, lines 19-34, where psychometric evaluation is described (references for PCA, the criterion for Bartlett's test and KMO, Kaiser's criterion, component loading cut off). Acceptable cut off scores for Cronbach's alpha coefficient should also be stated in the next paragraph (page 7, line 41), because some authors find Cronbach's alpha over 0.70 acceptable (for example Polit & Beck, 2004), but others (for example Streiner 2003) find that higher values (over 0.90 or so) reflect unnecessary duplication and point more redundancy than the homogeneity.

8. Most references are adequately chosen. The authors cite the papers of the countries in which PCS-S was validated. One additional psychometric evaluation of PCQ-S has been recently published (Psychometric testing of the Slovenian Person-centred Climate Questionnaire – staff version in Journal of Nursing Management). Including this paper and comparing results also with Slovenian version, would make this manuscript even more up-to-date. As already stated appropriate statistical literature could be included on page 7, lines 20-34. When describing the instrument, reference (19) is used for describing the English version (page 5, line 45), but (19) describes a PCQ-S in Swedish sample. Also, the English version of PCQ-S questionnaire has four subscales (Edvardsson et al. 2010), therefore it would be better to write "The original PCQ-S questionnaire consists of 14 items and has three subscales" (and add also more appropriate reference, Edvardsson et al. 2009). The rules for reference formatting are not fully followed in References (some Journal names are abbreviated some not).

9. Results address the research objective.

10. Results are presented clearly. It would be interesting to see how scores varied between hospitals. This would add knowledge about discriminatory capacity of the instrument and could be included also in discussion. In Results – Construct validity, page 8, line 30, authors have written that first component consists of five items (loadings between 0.58 and 0.83), but from Table 2, loadings are between 0.58 and 0.84.

11. The discussion and conclusion are justified by the results. The discussion largely repeats the findings. More in-depth discussion would improve this manuscript even more. Theoretical implications and implications for practice, further research and management could also be stated. In addition, in the Discussion section, page 10, line 8, authors have stated "In the Chinese PCQ-S, the ICC (0.94) and the Cronbach's alpha for total scale (0.94) was..." But in the previous paragraph (page 10, line 7) and in the Results section (page 8, line 45) authors have stated that Cronbach's alpha for total scale is 0.89. Cronbach's alpha of 0.94 for total scale is also stated in Abstract (page 2, line 49) and in Table 2. This should be checked and corrected through whole manuscript. In page 10, second paragraph (line 20-21), the authors have stated that English version consists of three subscales, this is not true, as the English version (Edvardsson et al. 2010) consists of four subscales. I presume this

	is a typing error, which should be corrected. Also, keep in mind, that an original version is Swedish version, what should be evident also in Discussion section. 12. Study limitations are discussed. Authors could add that content validity and criterion validity were not evaluated. 13. The supplementary STROBE checklist is not available. 14. It seems that manuscript is free from concerns over publication ethics. 15. Person-centeredness is mainly described in the Introduction section. As PCQ-S is developed to measure person-centred climate, describing the later, in more details would be useful for readers. Authors should also justify the choice of the PCQ-S in Introduction section, as there are many similar instruments.
--	--

VERSION 1 – AUTHOR RESPONSE

Reviewer: 1

Reviewer Name: Mariusz Panczyk, PhD

Institution and Country: Division of Teaching and Outcomes of Education, Faculty of Health Science, Medical University of Warsaw, POLAND

Please state any competing interests or state 'None declared': None declared

Please leave your comments for the authors below

The manuscript presented for evaluation is focused on cultural adaptation and psychometric validation of the Person-centred Climate Questionnaire – staff version (PCQ-S). The implementation of the idea of person-centred care helps to increase the quality of patient care and improves patient safety. The scale presented in the manuscript is a possible instrument that could be used to perform measurements of to what extent staff perceive different clinical settings or climates as being person-centred, an attribute often described as an essential component of quality health care. These are a few comments that would be helpful in improving the manuscript before the editors of the BMJ Open consider whether to publish it.

1. Introduction

The first paragraph reviewed the present demographic situation and outlined a picture of trends observed in Chinese society, hence providing a view of challenges faced by the Chinese healthcare system. The authors pointed to the introduction of the concept of person-centred care into the patient care system as an essential component of necessary reforms. This was supported by a number of available publications (e.g. McCormack B et al., Edvardsson D et al.) and seemed to be sufficient justification to move towards the implementation of the idea of person-centred care in China as well. Unfortunately, it seems that the authors provided insufficient justification for using the PCQ-S as the assessment tool. There was no information whether this was the only available questionnaire/scale relating to the person-centred care. In addition, it would be appropriate to devote a few words to theoretical assumptions that were the cornerstone of the original Swedish version of the PCQ-S and explain to what extent, according to the authors, these assumptions were accurate in relation to the Chinese culture. Is it possible that the idea of person-centred care is equally understood regardless of the specific nature and conditions in different countries?

Thank you for your very good comments. We have added some sentences to further explain the scale relating to the person-centred care and PCQ-S according to your comments. Please see Page 4, 2nd and 3rd paragraph.

2. Instrument

[page 5, line 56] "The English PCQ-S has previously been used and tested in hospital settings, and demonstrated to be a valid and reliable tool for assessing staff perceptions of the unit person-centredness [20]". Is there only one publication referring to the validation of the English-language version of the PCQ-S? Would it be advisable to mention psychometric properties of the original questionnaire (Swedish-language version), that is a prototype for the later English-language version?

There is only one publication referring to the validation of the English-language version of the PCQ-S. We have added some sentences to mention psychometric properties of the Swedish-language version. Please see Page 4, 2nd Paragraph.

3. Translation and cross-cultural adaptation of the PCQ-S

Despite the fact that the authors referred to two types of guidelines concerning the principles of psychometric validation, no reasons were provided why only the English-language version was used for validation. Good practices show that in the case of numerous language versions, a translation should be provided from different sources (different languages), and then a consensus version should be developed. This is all the more important because the original language version is Swedish, not English.

Because of the fact that the researcher who developed both the Swedish and the English versions of the PCQ-S is one of the co-authors and has been closely involved in this psychometric study, we therefore feel that we have taken the necessary steps to construct a valid cross-cultural adapted version of the PCQ-S.

Was the back-translated version consulted with the authors of the English-language version of the PCQ-S? Such consultation increases validity of the translation and allows for eliminating any inaccuracies from the new version regarding for instance the selection of expressions or translation of ambiguities.

The back-translated version was not consulted with the authors of the English-language version of the PCQ-S. We have added one sentence to explain it. Please see Page 5, last Paragraph.

[page 6, line 28] "A few wordings were adapted to the Chinese cultural setting (...)". Since the manuscript deals with cultural adaptation, it seems that the outcomes of this particular validation phase should be somehow made available to readers.

Done as what you suggested. Please see Page 10, Discussion section, 2nd Paragraph.

Did 10 persons participating in face-validity of the pre-final version subsequently take part in the study itself?

Ten persons participating in face-validity of the pre-final version did not subsequently take part in the study itself. The hospital they are from is not included in our study site. We have added one sentence to explain it. Please see Page 6, 1st Paragraph, last sentence.

4. Data collection

[page 7, line 7] "(...) completed questionnaires were anonymously collected on site." If the study was anonymous, how were the data obtained in the test phase linked with the data obtained in the retest phase? Both phases were separated by one week.

Each participant was assigned a number to indicate his or her identity, so they are anonymous. We have added sentences to explain it. Please see Page 7, 1st Paragraph.

5. Psychometric evaluation

PCA with varimax orthogonal rotation was used for the assessment of the construct validity. Assuming, however, that particular subscales may be strongly correlated, it would be advisable to use an oblique rotation method.

Thank you for your good comments! We have changed PCA with varimax orthogonal rotation as PCA with oblique rotation. Changing the rotation method did not affect the results estimated in varimax orthogonal rotation method. Please see Page 7, Psychometric evaluation section, 1st Paragraph.

In addition, would it not be appropriate to use a confirmatory factor analysis (CFA)? The validation of the original language version of PCQ-S demonstrated that there were three subscales. It would be useful to assess the consistency of the assumptions of such a model with the validation results of the Chinese version.

Done as what you suggested. Please see Page 7, Psychometric evaluation section, 1st Paragraph, and Page 9, Construct validity section, 1st Paragraph.

In addition, the authors provided no reference concerning the conditions applied for carrying out PCA (EFA?).

We have added two references concerning the conditions applied for carrying out PCA. Please see Page 7, Psychometric evaluation section, 1st and 2nd Paragraph. Please see Reference section, reference 32 and 33.

When assessing the internal consistency, the authors did not provide a satisfactory level of Cronbach's alpha coefficient.

We have provided an acceptable level of Cronbach's alpha coefficient. Please see Page 8, 1 Paragraph.

With respect to test-retest reliability, the intraclass correlation (ICC) was measured correctly. However, for the subscales scores, Bland Altman 95% limits of agreement of differences were determined, as a measure of the upper and lower limits of the differences between the scores on the two occasions of testing (see details: Giavarina D. Understanding Bland Altman analysis. *Biochem Med (Zagreb)*. 2015 Jun 5;25(2):141-51. doi: 10.11613/BM.2015.015).

Thank you for your good comments. We have discussed the method in the research group and have decided to keep our chosen method since it is a common method used in psychometric evaluations.

6. Results: Construct validity

The authors did not explain how the factor solution obtained in the study met the Kaiser's criterion. All that is known is that a 3-factor solution explained a total of 73.3% of the total variance. Did only the first three eigenvalues meet the criterion (≥ 1)?

Yes, only the first three components had eigenvalue greater than one, explaining 73.3% of the total variance. Therefore, the PCA resulted in a three-component rotated solution. We have revised the sentences to make it clearer. Please see Page 8, Construct validity section, last Paragraph.

7. Results: Reliability

[page 8, line 51] "(...) each item correlated adequately with the total score and thus that the scale is homogenous without any item being redundant". The EFA method needs to be used for the assessment of one-dimensionality of subscales (see details: Evaluating Psychometric Properties: Dimensionality and Reliability [In:] Furr, Mike. Scale construction and psychometrics for social and personality psychology. SAGE Publications Ltd, 2011).

Yes, we have used the explorative factor analyses (EFA) to obtain the three-component model. Please see Page 7, Psychometric evaluation section, 1st Paragraph.

8. Discussion

[page 9, line 28]: "In the Chinese PCQ-S, the ICC (0.97) and Cronbach's alpha for the total scale (0.94) were much higher than those recorded in Swedish (0.51 vs. 0.88) and English (0.75 vs. 0.89) versions, and the Cronbach's alpha for the total scale was also higher than the Norwegian version (0.92), (...)". It is important to recognize that the value of Cronbach's alpha coefficient is not only correlated with the internal consistency, but also with the sample size. Therefore, while presenting results in this area, it is important to bear in mind the comparability of groups with respect to the sample size in particular validation studies.

Thank you for your good comments. We have revised our sentences. Please see Page 11, 1st Paragraph.

[Page 10, line 45] With reference to the limitations of the study, it would also be worth mentioning that the questionnaire had been translated only from the secondary English version. Moreover, with respect to the psychometric assessment of the PCQ-S, convergent validity and discriminative validity were not taken into account. Additionally, the back-translated version was not validated, which, of course, would cause some technical difficulties due to the necessity to conduct the study among bilingual persons, but, at the same time, this would significantly increase confidence in the quality of cultural adaptation.

Done as what you suggested. Please see Page 11, 2nd Paragraph.

Summary

The manuscript presented for evaluation concerns an important issue of assessing the possibility of using the concept of person-centred care. This is important because the emerging negative demographic trends call for new solutions in patient care in numerous world countries, including China. The present PCQ-S validation outcomes may later contribute to an increase in the range of

applications of standardised tools used for planning changes and implementing the concept of person-centred care in Chinese healthcare centres and perhaps in other Asian countries as well. Making improvements accordingly will raise the quality of the texts itself and provide readers with a better understanding of the principles of a proper validation of scales and questionnaires from a different cultural background.

Reviewer: 2

Reviewer Name: Rie Chiba

Institution and Country: University of Hyogo, Japan

Please state any competing interests or state 'None declared': None declared

Please leave your comments for the authors below

Thank you for the opportunity to review the paper entitled "Psychometric evaluation of the Chinese version Person-centred Climate Questionnaire – staff."

In this paper, the authors examined its reliability and validity of the scale in the palliative ward settings. The article is well written and interesting. I think it is valuable to be published in the journal. I hope some points described below would be the tips to refine the article.

1. Background

In this paper, the authors conducted the survey only in palliative wards. Thus the explanation about general circumstances of palliative care in China, as well as the significance about the assessment of person-centered climate among the staff in palliative words may be needed.

Done as what you suggested. Please see Page 3, Introduction section, 1st Paragraph.

2. Sample and Participants

Though it is mentioned that only oral information about the study was provided, did they receive writing information? If so, the explanation may be added.

They received both oral and written information about the study. We have added the explanation to the paper. Please see Page 6, Sample and participants section.

3. Sample and Participants

The response rate (90%) seems considerably high. And it also seems too impeccable that there was no missing data among 163 participants at two time points. Thus the authors may want to describe about them including whether the explanation about such as the non-participants would not be disadvantaged, or participants were allowed to withdraw from the study at any time without prejudice were provided or not.

Done as what you suggested. Please see Page 6, Sample and participants section.

4. Sample and Participants

Additional explanation about the timeframe of the test-retest may be desirable because there might be some participants who could not answer the retest survey just after a week from the baseline survey,

because of their work schedule.

Done as what you suggested. Please see page 7, Data collection section, last sentence.

5. Psychometric evaluation

The authors stated that they chose varimax rotation to ensure independence of the items. But I wonder the correlations among each items might be hypothesized. Thus more convincing explanation including the description about the rotations in the earlier studies may be desirable if the rotation was adopted.

We have changed PCA with varimax orthogonal rotation as PCA with oblique rotation according to another reviewer's suggestion. Changing the rotation method did not affect the results estimated in varimax orthogonal rotation method. Please see Page 7, Psychometric evaluation section, 1st Paragraph.

6. Psychometric evaluation

I wonder why the criterion-related validity was not examined in this study. If such data was obtained, please consider to show the result of the analyses.

Unfortunately, such data was not obtained in our study. We have mentioned it as one of our limitations. Please see Page 11, 2nd Paragraph.

7. Reliability

Regarding test-retest reliability, the authors may want to analyze using weighted kappa which would show more rigorous result compared to the Pearson's correlation.

We did not use weighted kappa as to enable cross-publication comparisons where most previous publications have used Pearson's correlation.

8. Discussion

The first paragraph except for the last sentence may be better to be deleted or removed to Introduction section. In addition, the second section also seems duplicate the Introduction. Briefer summary of the result of the current study seems preferable.

We have deleted some sentences according to your good suggestion. Please see Page 10, Discussion section, 1st Paragraph.

9. Discussion

In relation to the third paragraph, more concrete description of the factorial constructions in the earlier studies and profound discussion may add some implication of the study, including the difference of the culture between China and Western countries.

Done as what you suggested. Please see Page 10, 3rd paragraph.

10. Limitation

The author may want to add the limitation that criterion-related validity was not examined in this study.

Done as what you suggested. Please see Page 11, 2nd paragraph.

Reviewer: 3

Reviewer Name: Dominika Vrbnjak

Institution and Country: University of Maribor Faculty of Health Sciences

Please state any competing interests or state 'None declared': None declared

Please leave your comments for the authors below

Thank you for an interesting manuscript, it contains useful content. The manuscript is well written and adds to the international literature on measuring person-centeredness of environments. However, there are some issues to be considered, that might improve the manuscript further.

1. Study objective is clearly defined.

Thank you.

2. Abstract is balanced and complete. The sample included health care staff only from palliative care, this should be evident also from abstract and abstract summary.

Done as what you suggested. Please see Abstract, Objectives section.

3. The study design is adequate to address the study objective.

Thank you.

4. Methods are sufficiently described to allow the study to be repeated. But, no sample size calculation is available. KMO was performed to measure the sampling adequacy, however, this was done after distributing the questionnaires. Justifying the rationale for sample size and convenience sampling is needed. Justifying the choice of three hospitals would also be appropriate.

Done as what you suggested. Please see Page 6, Sample and participants section.

Also, it would be useful if a more detailed description of what all staff on duty is meant (morning shift only?).

Done as what you suggested. Please see Page 6, Sample and participants section.

In Data collection section, page 7, line 3-5 authors describe collecting demographic data. In addition to listed age, sex, level of education and duration of work experience, the authors have also collected data about ethnicity and health care staff position, this should also be listed here.

Done as what you suggested. Please see Page 7, Data collection section, 1st Paragraph.

5. Ethics approval is stated; however, there could be more in-depth description of ethical issues, especially as test-retest was done. Authors could explain how anonymity was assured and describe coding of the questionnaire if this was done.

Done as what you suggested. Please see Page 7, 1st Paragraph.

6. Outcomes are clearly defined.

Thank you.

7. Statistical methods adequately match the study. Appropriate statistical references could be included on page 7, lines 19-34, where psychometric evaluation is described (references for PCA, the criterion for Bartlett's test and KMO, Kaiser's criterion, component loading cut off).

We have added two references. Please see reference section, reference 32 and 33.

Acceptable cut off scores for Cronbach's alpha coefficient should also be stated in the next paragraph (page 7, line 41), because some authors find Cronbach's alpha over 0.70 acceptable (for example Polit & Beck, 2004), but others (for example Streiner 2003) find that higher values (over 0.90 or so) reflect unnecessary duplication and point more redundancy than the homogeneity.

Done as what you suggested. Please see Page 7, last paragraph, last sentence, and Page 8, 1st Paragraph.

8. Most references are adequately chosen. The authors cite the papers of the countries in which PCS-S was validated. One additional psychometric evaluation of PCQ-S has been recently published (Psychometric testing of the Slovenian Person-centred Climate Questionnaire – staff version in Journal of Nursing Management). Including this paper and comparing results also with Slovenian version, would make this manuscript even more up-to-date. As already stated appropriate statistical literature could be included on page 7, lines 20-34. When describing the instrument, reference (19) is used for describing the English version (page 5, line 45), but (19) describes a PCQ-S in Swedish sample. Also, the English version of PCQ-S questionnaire has four subscales (Edvardsson et al. 2010), therefore it would be better to write "The original PCQ-S questionnaire consists of 14 items and has three subscales" (and add also more appropriate reference, Edvardsson et al. 2009). The rules for reference formatting are not fully followed in References (some Journal names are abbreviated some not).

We have added this new publication and also compared results with Slovenian version. Please see Reference section, reference 26, Page 4, last Paragraph, and Page 10, 3rd Paragraph. Furthermore, we adjust the order of references in Reference section to keep consistent with the cited one. Please see Reference section. We also correct the error in this paper. Please see Page 10, 3rd Paragraph.

The reference formatting is also corrected. Please see Reference section.

9. Results address the research objective.

Thank you.

10. Results are presented clearly. It would be interesting to see how scores varied between hospitals. This would add knowledge about discriminatory capacity of the instrument and could be included also in discussion.

Unfortunately, the hospital information was not included in our data, so we can't analyze how scores varied between hospitals.

In Results – Construct validity, page 8, line 30, authors have written that first component consists of five items (loadings between 0.58 and 0.83), but from Table 2, loadings are between 0.58 and 0.84.

Thank you very much for your comments. We have corrected the mistakes. Please see Page 8, Construct validity section.

11. The discussion and conclusion are justified by the results. The discussion largely repeats the findings. More in-depth discussion would improve this manuscript even more. Theoretical implications and implications for practice, further research and management could also be stated. In addition, in the Discussion section, page 10, line 8, authors have stated "In the Chinese PCQ-S, the ICC (0.94) and the Cronbach's alpha for total scale (0.94) was..." But in the previous paragraph (page 10, line 7) and in the Results section (page 8, line 45) authors have stated that Cronbach's alpha for total scale is 0.89. Cronbach's alpha of 0.94 for total scale is also stated in Abstract (page 2, line 49) and in Table 2. This should be checked and corrected through whole manuscript. In page 10, second paragraph (line 20-21), the authors have stated that English version consists of three subscales, this is not true, as the English version (Edvardsson et al. 2010) consists of four subscales. I presume this is a typing error, which should be corrected. Also, keep in mind, that an original version is Swedish version, what should be evident also in Discussion section.

Thank you very much for your good comments. We have removed some sentences that duplicate the Introduction. Please see Discussion section, Page 10, 1st Paragraph. Please also see Discussion section, Page 10, 2nd Paragraph, and Page 11, Conclusion section.

We also corrected the errors in this paper. Please see Abstract section, and Discussion section, Page 10, 3rd Paragraph.

12. Study limitations are discussed. Authors could add that content validity and criterion validity were not evaluated.

Done as what you suggested. Please see Page 11, 2nd paragraph.

13. The supplementary STROBE checklist is not available.

Done as what you suggested. We have provided the supplementary STROBE checklist.

14. It seems that manuscript is free from concerns over publication ethics.

Please see Page 12, Consent for publication section.

15. Person-centeredness is mainly described in the Introduction section. As PCQ-S is developed to measure person-centred climate, describing the later, in more details would be useful for readers.

Authors should also justify the choice of the PCQ-S in Introduction section, as there are many similar instruments.

Done as what you suggested. Please see Page 4, 2nd and 3rd Paragraph.

VERSION 2 – REVIEW

REVIEWER	Mariusz Panczyk, PhD Medical University of Warsaw, POLAND
REVIEW RETURNED	04-Jul-2017

GENERAL COMMENTS	I am more or less happy now. One tiny revision. I still think that test-retest reliability (absolute stability) should be measured by calculating the weighted kappa coefficient. To establish a correct test-retest analysis, the assumption of equal means in two measurements was checked using the t test.
--

REVIEWER	Rie Chiba University of Hyogo Japan
REVIEW RETURNED	02-Jul-2017

GENERAL COMMENTS	Thank you for the opportunity to review the revised manuscript entitled "Psychometric evaluation of the Chinese version of the Person-centred Climate Questionnaire for staff." Most of points I noted earlier were revised adequately. The matter regarding test-retest reliability that I suggested to use weighted kappa was not considered. But generally, the authors made changes to make the manuscript more valuable.
--

REVIEWER	Dominika Vrbnjak University of Maribor Faculty of Health Sciences
REVIEW RETURNED	23-Jun-2017

GENERAL COMMENTS	The reviewer's comments appear to be addressed. However, authors have stated that (27) is only one publication referring to the validation of the English-language version of the PCQ-S. This is not true, as the Slovenian version was translated from English-language version. This should be corrected in the text: This instrument has been translated from Swedish into Norwegian [26], English [27] and Slovenian [28].
---

VERSION 2 – AUTHOR RESPONSE

Reviewer: 3

Reviewer Name: Dominika Vrbnjak

Institution and Country: University of Maribor Faculty of Health Sciences

Please state any competing interests or state 'None declared': None declared

The reviewer's comments appear to be addressed.

However, authors have stated that (27) is only one publication referring to the validation of the English-language version of the PCQ-S. This is not true, as the Slovenian version was translated from

English-language version. This should be corrected in the text: This instrument has been translated from Swedish into Norwegian [26], English [27] and Slovenian [28].

Thank you for your very good comments. We have revised this sentence according to your comments. Please see Page 4, last paragraph.

Reviewer: 2

Reviewer Name: Rie Chiba

Institution and Country: University of Hyogo, Japan

Please state any competing interests or state 'None declared': None declared

Please leave your comments for the authors below

Thank you for the opportunity to review the revised manuscript entitled "Psychometric evaluation of the Chinese version of the Person-centred Climate Questionnaire for staff."

Most of points I noted earlier were revised adequately. The matter regarding test-retest reliability that I suggested to use weighted kappa was not considered. But generally, the authors made changes to make the manuscript more valuable.

Done as what you suggested. Weighted Kappa has been used to measure test-retest reliability. We also keep the Pearson correlation coefficient (r) as to enable cross-publication comparisons where most previous publications have used Pearson's correlation. Please see Method section page 8, 1st Paragraph; Results section page 9, last Paragraph; Table 4; Abstract under Design page 1 and under Result page 2.

Reviewer: 1

Reviewer Name: Mariusz Panczyk, PhD

Institution and Country: Medical University of Warsaw, POLAND

Please state any competing interests or state 'None declared': None declared

Please leave your comments for the authors below

I am more or less happy now. One tiny revision. I still think that test-retest reliability (absolute stability) should be measured by calculating the weighted kappa coefficient. To establish a correct test-retest analysis, the assumption of equal means in two measurements was checked using the t test.

Done as what you suggested. Weighted Kappa has been used to measure test-retest reliability. Please see Method section page 8, 1st Paragraph; Results section page 9, last Paragraph; Table 4; Abstract under Design page 1 and under Result page 2.